# LEARNING TO OPTIMIZE NEURAL NETS

## ABSTRACT

Learning to Optimize (Li & Malik, 2016) is a recently proposed framework for learning optimization algorithms using reinforcement learning. In this paper, we explore learning an optimization algorithm for training shallow neural nets. Such high-dimensional stochastic optimization problems present interesting challenges for existing reinforcement learning algorithms. We develop an extension that is suited to learning optimization algorithms in this setting and demonstrate that the learned optimization algorithm consistently outperforms other known optimization algorithms even on unseen tasks and is robust to changes in stochasticity of gradients and the neural net architecture. More specifically, we show that an optimization algorithm trained with the proposed method on the problem of training a neural net on MNIST generalizes to the problems of training neural nets on the Toronto Faces Dataset, CIFAR-10 and CIFAR-100.

## 1 INTRODUCTION

Machine learning is centred on the philosophy that learning patterns automatically from data is generally better than meticulously crafting rules by hand. This data-driven approach has delivered: today, machine learning techniques can be found in a wide range of application areas, both in AI and beyond. Yet, there is one domain that has conspicuously been left untouched by machine learning: the design of tools that power machine learning itself.

One of the most widely used tools in machine learning is optimization algorithms. We have grown accustomed to seeing an optimization algorithm as a black box that takes in a model that we design and the data that we collect and outputs the optimal model parameters. The optimization algorithm itself largely stays static: its design is reserved for human experts, who must toil through many rounds of theoretical analysis and empirical validation to devise a better optimization algorithm. Given this state of affairs, perhaps it is time for us to start practicing what we preach and learn how to learn.

Recently, Li & Malik (2016) and Andrychowicz et al. (2016) introduced two different frameworks for learning optimization algorithms. Whereas Andrychowicz et al. (2016) focuses on learning an optimization algorithm for training models on a particular task, Li & Malik (2016) sets a more ambitious objective of learning an optimization algorithm for training models that is task-independent. We study the latter paradigm in this paper and develop a method for learning an optimization algorithm for high-dimensional stochastic optimization problems, like the problem of training shallow neural nets.

Under the "Learning to Optimize" framework proposed by Li & Malik (2016), the problem of learning an optimization algorithm is formulated as a reinforcement learning problem. We consider the general structure of an unconstrained continuous optimization algorithm, as shown in Algorithm 1. In each iteration, the algorithm takes a step $\Delta x$ and uses it to update the current iterate $x^{(i)}$. In hand-engineered optimization algorithms, $\Delta x$ is computed using some fixed formula $\phi$ that depends on the objective function, the current iterate and past iterates. Often, it is simply a function of the current and past gradients.

Different choices of $\phi$ yield different optimization algorithms and so each optimization algorithm is essentially characterized by its update formula $\phi$. Hence, by learning $\phi$, we can learn an optimization algorithm. Li & Malik (2016) observed that an optimization algorithm can be viewed as a Markov decision process (MDP), where the state includes the current iterate, the action is the step vector $\Delta x$

---

**Algorithm 1** General structure of optimization algorithms

---

**Require:** Objective function $f$
   $x^{(0)} \leftarrow$ random point in the domain of $f$
   **for** $i = 1, 2, \ldots$ **do**
      $\Delta x \leftarrow \phi(f, \{x^{(0)}, \ldots, x^{(i-1)}\})$
      **if** stopping condition is met **then**
         **return** $x^{(i-1)}$
      **end if**
      $x^{(i)} \leftarrow x^{(i-1)} + \Delta x$
   **end for**

---

and the policy is the update formula $\phi$. Hence, the problem of learning $\phi$ simply reduces to a policy search problem.

In this paper, we build on the method proposed in (Li & Malik, 2016) and develop an extension that is suited to learning optimization algorithms for high-dimensional stochastic problems. We use it to learn an optimization algorithm for training shallow neural nets and show that it outperforms popular hand-engineered optimization algorithms like ADAM (Kingma & Ba, 2014), AdaGrad (Duchi et al., 2011) and RMSprop (Tieleman & Hinton, 2012) and an optimization algorithm learned using the supervised learning method proposed in (Andrychowicz et al., 2016). Furthermore, we demonstrate that our optimization algorithm learned from the experience of training on MNIST generalizes to training on other datasets that have very dissimilar statistics, like the Toronto Faces Dataset, CIFAR-10 and CIFAR-100.

## 2 RELATED WORK

The line of work on learning optimization algorithms is fairly recent. Li & Malik (2016) and Andrychowicz et al. (2016) were the first to propose learning general optimization algorithms. Li & Malik (2016) explored learning task-independent optimization algorithms and used reinforcement learning to learn the optimization algorithm, while Andrychowicz et al. (2016) investigated learning task-dependent optimization algorithms and used supervised learning.

In the special case where objective functions that the optimization algorithm is trained on are loss functions for training other models, these methods can be used for "learning to learn" or "meta-learning". While these terms have appeared from time to time in the literature (Baxter et al., 1995; Vilalta & Drissi, 2002; Brazdil et al., 2008; Thrun & Pratt, 2012), they have been used by different authors to refer to disparate methods with different purposes. These methods all share the objective of learning some form of meta-knowledge about learning, but differ in the type of meta-knowledge they aim to learn. We can divide the various methods into the following three categories.

### 2.1 LEARNING WHAT TO LEARN

Methods in this category Thrun & Pratt (2012) aim to learn what parameter values of the base-level learner are useful across a family of related tasks. The meta-knowledge captures commonalities shared by tasks in the family, which enables learning on a new task from the family to be performed more quickly. Most early methods fall into this category; this line of work has blossomed into an area that has later become known as transfer learning and multi-task learning.

### 2.2 LEARNING WHICH MODEL TO LEARN

Methods in this category Brazdil et al. (2008) aim to learn which base-level learner achieves the best performance on a task. The meta-knowledge captures correlations between different tasks and the performance of different base-level learners on those tasks. One challenge under this setting is to decide on a parameterization of the space of base-level learners that is both rich enough to be capable of representing disparate base-level learners and compact enough to permit tractable search over this space. Brazdil et al. (2003) proposes a nonparametric representation and stores examples of different base-level learners in a database, whereas Schmidhuber (2004) proposes representing base-

level learners as general-purpose programs. The former has limited representation power, while the latter makes search and learning in the space of base-level learners intractable. Hochreiter et al. (2001) views the (online) training procedure of any base-learner as a black box function that maps a sequence of training examples to a sequence of predictions and models it as a recurrent neural net. Under this formulation, meta-training reduces to training the recurrent net, and the base-level learner is encoded in the memory state of the recurrent net.

Hyperparameter optimization can be seen as another example of methods in this category. The space of base-level learners to search over is parameterized by a predefined set of hyperparameters. Unlike the methods above, multiple trials with different hyperparameter settings on the same task are permitted, and so generalization across tasks is not required. The discovered hyperparameters are generally specific to the task at hand and hyperparameter optimization must be rerun for new tasks. Various kinds of methods have been proposed, such those based on Bayesian optimization (Hutter et al., 2011; Bergstra et al., 2011; Snoek et al., 2012; Swersky et al., 2013; Feurer et al., 2015), random search (Bergstra & Bengio, 2012) and gradient-based optimization (Bengio, 2000; Domke, 2012; Maclaurin et al., 2015).

## 2.3 LEARNING HOW TO LEARN

Methods in this category aim to learn a good algorithm for training a base-level learner. Unlike methods in the previous categories, the goal is not to learn about the *outcome* of learning, but rather the *process* of learning. The meta-knowledge captures commonalities in the behaviours of learning algorithms that achieve good performance. The base-level learner and the task are given by the user, so the learned algorithm must generalize across base-level learners and tasks. Since learning in most cases is equivalent to optimizing some objective function, learning a learning algorithm often reduces to learning an optimization algorithm. This problem was explored in (Li & Malik, 2016) and (Andrychowicz et al., 2016). Closely related is (Bengio et al., 1991), which learns a Hebb-like synaptic learning rule that does not depend on the objective function, which does not allow for generalization to different objective functions.

Various work has explored learning how to adjust the hyperparameters of hand-engineered optimization algorithms, like the step size (Hansen, 2016; Daniel et al., 2016; Fu et al., 2016) or the damping factor in the Levenberg-Marquardt algorithm (Ruvolo et al., 2009). Related to this line of work is stochastic meta-descent (Bray et al., 2004), which derives a rule for adjusting the step size analytically. A different line of work (Gregor & LeCun, 2010; Sprechmann et al., 2013) parameterizes intermediate operands of special-purpose solvers for a class of optimization problems that arise in sparse coding and learns them using supervised learning.

# 3 LEARNING TO OPTIMIZE

## 3.1 SETTING

In the "Learning to Optimize" framework, we are given a set of training objective functions $f_1, \ldots, f_n$ drawn from some distribution $\mathcal{F}$. An optimization algorithm $\mathcal{P}$ takes an objective function $f$ and an initial iterate $x^{(0)}$ as input and produces a sequence of iterates $x^{(1)}, \ldots, x^{(T)}$, where $x^{(T)}$ is the solution found by the optimizer. We are also given a distribution $\mathcal{D}$ that generates the initial iterate $x^{(0)}$ and a meta-loss $\mathcal{L}$, which takes an objective function $f$ and a sequence of iterates $x^{(1)}, \ldots, x^{(T)}$ produced by an optimization algorithm as input and outputs a scalar that measures the quality of the iterates. The goal is to learn an optimization algorithm $\mathcal{P}^*$ such that $\mathbb{E}_{f \sim \mathcal{F}, x^{(0)} \sim \mathcal{D}} \left[ \mathcal{L}(f, \mathcal{P}^*(f, x^{(0)})) \right]$ is minimized. The meta-loss is chosen to penalize optimization algorithms that exhibit behaviours we find undesirable, like slow convergence or excessive oscillations. Assuming we would like to learn an algorithm that minimizes the objective function it is given, a good choice of meta-loss would then simply be $\sum_{i=1}^{T} f(x^{(i)})$, which is equivalent to cumulative regret and can be interpreted as the area under the curve of objective values over time.

The objective functions $f_1, \ldots, f_n$ may correspond to loss functions for training base-level learners, in which case the algorithm that learns the optimization algorithm can be viewed as a meta-learner. In this setting, each objective function is the loss function for training a particular base-learner on a particular task, and so the set of training objective functions can be loss functions for training a

base-learner or a family of base-learners on different tasks. At test time, the learned optimization algorithm is evaluated on unseen objective functions, which correspond to loss functions for training base-learners on new tasks, which may be completely unrelated to tasks used for training the optimization algorithm. Therefore, the learned optimization algorithm must not learn anything about the tasks used for training. Instead, the goal is to learn an optimization algorithm that can exploit the geometric structure of the error surface induced by the base-learners. For example, if the base-level model is a neural net with ReLU activation units, the optimization algorithm should hopefully learn to leverage the piecewise linearity of the model. Hence, there is a clear division of responsibilities between the meta-learner and base-learners. The knowledge learned at the meta-level should be pertinent for all tasks, whereas the knowledge learned at the base-level should be task-specific. The meta-learner should therefore generalize across tasks, whereas the base-learner should generalize across instances.

## 3.2 RL PRELIMINARIES

The goal of reinforcement learning is to learn to interact with an environment in a way that minimizes cumulative costs that are expected to be incurred over time. The environment is formalized as a partially observable Markov decision process (POMDP)[1], which is defined by the tuple $(\mathcal{S}, \mathcal{O}, \mathcal{A}, p_i, p, p_o, c, T)$, where $\mathcal{S} \subseteq \mathbb{R}^D$ is the set of states, $\mathcal{O} \subseteq \mathbb{R}^{D'}$ is the set of observations, $\mathcal{A} \subseteq \mathbb{R}^d$ is the set of actions, $p_i(s_0)$ is the probability density over initial states $s_0$, $p(s_{t+1}|s_t, a_t)$ is the probability density over the subsequent state $s_{t+1}$ given the current state $s_t$ and action $a_t$, $p_o(o_t|s_t)$ is the probability density over the current observation $o_t$ given the current state $s_t$, $c : \mathcal{S} \to \mathbb{R}$ is a function that assigns a cost to each state and $T$ is the time horizon. Often, the probability densities $p$ and $p_o$ are unknown and not given to the learning algorithm.

A policy $\pi(a_t|o_t, t)$ is a conditional probability density over actions $a_t$ given the current observation $o_t$ and time step $t$. When a policy is independent of $t$, it is known as a stationary policy. The goal of the reinforcement learning algorithm is to learn a policy $\pi^*$ that minimizes the total expected cost over time. More precisely,

$$\pi^* = \arg\min_\pi \mathbb{E}_{s_0, a_0, s_1, \dots, s_T} \left[ \sum_{t=0}^{T} c(s_t) \right],$$

where the expectation is taken with respect to the joint distribution over the sequence of states and actions, often referred to as a trajectory, which has the density

$$q(s_0, a_0, s_1, \dots, s_T) = \int_{o_0, \dots, o_T} p_i(s_0) p_o(o_0|s_0)$$
$$\prod_{t=0}^{T-1} \pi(a_t|o_t, t) p(s_{t+1}|s_t, a_t) p_o(o_{t+1}|s_{t+1}).$$

To make learning tractable, $\pi$ is often constrained to lie in a parameterized family. A common assumption is that $\pi(a_t|o_t, t) = \mathcal{N}(\mu^\pi(o_t), \Sigma^\pi(o_t))$, where $\mathcal{N}(\mu, \Sigma)$ denotes the density of a Gaussian with mean $\mu$ and covariance $\Sigma$. The functions $\mu^\pi(\cdot)$ and possibly $\Sigma^\pi(\cdot)$ are modelled using function approximators, whose parameters are learned.

## 3.3 FORMULATION

In our setting, the state $s_t$ consists of the current iterate $x^{(t)}$ and features $\Phi(\cdot)$ that depend on the history of iterates $x^{(1)}, \dots, x^{(t)}$, (noisy) gradients $\nabla \hat{f}(x^{(1)}), \dots, \nabla \hat{f}(x^{(t)})$ and (noisy) objective values $\hat{f}(x^{(1)}), \dots, \hat{f}(x^{(t)})$. The action $a_t$ is the step $\Delta x$ that will be used to update the iterate. The observation $o_t$ excludes $x^{(t)}$ and consists of features $\Psi(\cdot)$ that depend on the iterates, gradient and objective values from recent iterations, and the previous memory state of the learned optimization algorithm, which takes the form of a recurrent neural net. This memory state can be viewed as a statistic of the previous observations that is learned jointly with the policy.

---

[1]What is described is an undiscounted finite-horizon POMDP with continuous state, observation and action spaces.

Under this formulation, the initial probability density $p_i$ captures how the initial iterate, gradient and objective value tend to be distributed. The transition probability density $p$ captures the how the gradient and objective value are likely to change given the step that is taken currently; in other words, it encodes the local geometry of the training objective functions. Assuming the goal is to learn an optimization algorithm that minimizes the objective function, the cost $c$ of a state $s_t = \left(x^{(t)}, \Phi\left(\cdot\right)\right)^T$ is simply the true objective value $f(x^{(t)})$.

Any particular policy $\pi\left(a_t \mid o_t, t\right)$, which generates $a_t = \Delta x$ at every time step, corresponds to a particular (noisy) update formula $\phi$, and therefore a particular (noisy) optimization algorithm. Therefore, learning an optimization algorithm simply reduces to searching for the optimal policy.

The mean of the policy is modelled as a recurrent neural net fragment that corresponds to a single time step, which takes the observation features $\Psi(\cdot)$ and the previous memory state as input and outputs the step to take.

### 3.4 GUIDED POLICY SEARCH

The reinforcement learning method we use is guided policy search (GPS) (Levine et al., 2015), which is a policy search method designed for searching over large classes of expressive non-linear policies in continuous state and action spaces. It maintains two policies, $\psi$ and $\pi$, where the former lies in a time-varying linear policy class in which the optimal policy can found in closed form, and the latter lies in a stationary non-linear policy class in which policy optimization is challenging. In each iteration, it performs policy optimization on $\psi$, and uses the resulting policy as supervision to train $\pi$.

More precisely, GPS solves the following constrained optimization problem:

$$\min_{\theta, \eta} \mathbb{E}_\psi \left[\sum_{t=0}^{T} c(s_t)\right] \text{ s.t. } \psi\left(a_t \mid s_t, t; \eta\right) = \pi\left(a_t \mid s_t; \theta\right) \; \forall a_t, s_t, t$$

where $\eta$ and $\theta$ denote the parameters of $\psi$ and $\pi$ respectively, $\mathbb{E}_\rho\left[\cdot\right]$ denotes the expectation taken with respect to the trajectory induced by a policy $\rho$ and $\pi\left(a_t \mid s_t; \theta\right) := \int_{o_t} \pi\left(a_t \mid o_t; \theta\right) p_o\left(o_t \mid s_t\right)^2$.

Since there are an infinite number of equality constraints, the problem is relaxed by enforcing equality on the mean actions taken by $\psi$ and $\pi$ at every time step[3]. So, the problem becomes:

$$\min_{\theta, \eta} \mathbb{E}_\psi \left[\sum_{t=0}^{T} c(s_t)\right] \text{ s.t. } \mathbb{E}_\psi\left[a_t\right] = \mathbb{E}_\psi\left[\mathbb{E}_\pi\left[a_t \mid s_t\right]\right] \; \forall t$$

This problem is solved using Bregman ADMM (Wang & Banerjee, 2014), which performs the following updates in each iteration:

$$\eta \leftarrow \arg\min_\eta \sum_{t=0}^{T} \mathbb{E}_\psi \left[c(s_t) - \lambda_t^T a_t\right] + \nu_t D_t\left(\eta, \theta\right)$$

$$\theta \leftarrow \arg\min_\theta \sum_{t=0}^{T} \lambda_t^T \mathbb{E}_\psi\left[\mathbb{E}_\pi\left[a_t \mid s_t\right]\right] + \nu_t D_t\left(\theta, \eta\right)$$

$$\lambda_t \leftarrow \lambda_t + \alpha\nu_t\left(\mathbb{E}_\psi\left[\mathbb{E}_\pi\left[a_t \mid s_t\right]\right] - \mathbb{E}_\psi\left[a_t\right]\right) \; \forall t,$$

where $D_t\left(\theta, \eta\right) := \mathbb{E}_\psi\left[D_{KL}\left(\pi\left(a_t \mid s_t; \theta\right) \| \psi\left(a_t \mid s_t, t; \eta\right)\right)\right]$ and $D_t\left(\eta, \theta\right) := \mathbb{E}_\psi\left[D_{KL}\left(\psi\left(a_t \mid s_t, t; \eta\right) \| \pi\left(a_t \mid s_t; \theta\right)\right)\right]$.

The algorithm assumes that $\psi\left(a_t \mid s_t, t; \eta\right) = \mathcal{N}\left(K_t s_t + k_t, G_t\right)$, where $\eta := (K_t, k_t, G_t)_{t=1}^{T}$ and $\pi\left(a_t \mid o_t; \theta\right) = \mathcal{N}\left(\mu_\omega^\pi(o_t), \Sigma^\pi\right)$, where $\theta := (\omega, \Sigma^\pi)$ and $\mu_\omega^\pi(\cdot)$ can be an arbitrary function that is typically modelled using a nonlinear function approximator like a neural net.

---

[2]In practice, the explicit form of the observation probability $p_o$ is usually not known or the integral may be intractable to compute. So, a linear Gaussian model is fitted to samples of $s_t$ and $a_t$ and used in place of the true $\pi\left(a_t \mid s_t; \theta\right)$ where necessary.

[3]Though the Bregman divergence penalty is applied to the original probability distributions over $a_t$.

At the start of each iteration, the algorithm constructs a model of the transition probability density $\tilde{p}\left(s_{t+1}|s_t, a_t, t; \zeta\right) = \mathcal{N}(A_t s_t + B_t a_t + c_t, F_t)$, where $\zeta := (A_t, B_t, c_t, F_t)_{t=1}^T$ is fitted to samples of $s_t$ drawn from the trajectory induced by $\psi$, which essentially amounts to a local linearization of the true transition probability $p\left(s_{t+1}|s_t, a_t, t\right)$. We will use $\mathbb{E}_{\tilde{\psi}}\left[\cdot\right]$ to denote expectation taken with respect to the trajectory induced by $\psi$ under the modelled transition probability $\tilde{p}$. Additionally, the algorithm fits local quadratic approximations to $c(s_t)$ around samples of $s_t$ drawn from the trajectory induced by $\psi$ so that $c(s_t) \approx \tilde{c}(s_t) := \frac{1}{2} s_t^T C_t s_t + d_t^T s_t + h_t$ for $s_t$'s that are near the samples.

With these assumptions, the subproblem that needs to be solved to update $\eta = (K_t, k_t, G_t)_{t=1}^T$ becomes:

$$\min_\eta \sum_{t=0}^T \mathbb{E}_{\tilde{\psi}}\left[\tilde{c}(s_t) - \lambda_t^T a_t\right] + \nu_t D_t\left(\eta, \theta\right)$$

$$\text{s.t. } \sum_{t=0}^T \mathbb{E}_{\tilde{\psi}}\left[D_{KL}\left(\psi\left(a_t|s_t, t; \eta\right)\| \psi\left(a_t|s_t, t; \eta'\right)\right)\right] \leq \epsilon,$$

where $\eta'$ denotes the old $\eta$ from the previous iteration. Because $\tilde{p}$ and $\tilde{c}$ are only valid locally around the trajectory induced by $\psi$, the constraint is added to limit the amount by which $\eta$ is updated. It turns out that the unconstrained problem can be solved in closed form using a dynamic programming algorithm known as linear-quadratic-Gaussian (LQG) regulator in time linear in the time horizon $T$ and cubic in the dimensionality of the state space $D$. The constrained problem is solved using dual gradient descent, which uses LQG as a subroutine to solve for the primal variables in each iteration and increments the dual variable on the constraint until it is satisfied.

Updating $\theta$ is straightforward, since expectations taken with respect to the trajectory induced by $\pi$ are always conditioned on $s_t$ and all outer expectations over $s_t$ are taken with respect to the trajectory induced by $\psi$. Therefore, $\pi$ is essentially decoupled from the transition probability $p\left(s_{t+1}|s_t, a_t, t\right)$ and so its parameters can be updated without affecting the distribution of $s_t$'s. The subproblem that needs to be solved to update $\theta$ therefore amounts to a standard supervised learning problem.

Since $\psi\left(a_t|s_t, t; \eta\right)$ and $\pi\left(a_t|s_t; \theta\right)$ are Gaussian, $D_t\left(\theta, \eta\right)$ can be computed analytically. More concretely, if we assume $\Sigma^\pi$ to be fixed for simplicity, the subproblem that is solved for updating $\theta = (\omega, \Sigma^\pi)$ is:

$$\min_\theta \mathbb{E}_\psi\left[\sum_{t=0}^T \lambda_t^T \mu_\omega^\pi(o_t) + \frac{\nu_t}{2}\left(\operatorname{tr}\left(G_t^{-1}\Sigma^\pi\right) - \log|\Sigma^\pi|\right)\right.$$

$$\left. + \frac{\nu_t}{2}\left(\mu_\omega^\pi(o_t) - \mathbb{E}_\psi\left[a_t|s_t, t\right]\right)^T G_t^{-1}\left(\mu_\omega^\pi(o_t) - \mathbb{E}_\psi\left[a_t|s_t, t\right]\right)\right]$$

Note that the last term is the squared Mahalanobis distance between the mean actions of $\psi$ and $\pi$ at time step $t$, which is intuitive as we would like to encourage $\pi$ to match $\psi$.

### 3.5 CONVOLUTIONAL GPS

The problem of learning high-dimensional optimization algorithms presents challenges for reinforcement learning algorithms due to high dimensionality of the state and action spaces. For example, in the case of GPS, because the running time of LQG is cubic in dimensionality of the state space, performing policy search even in the simple class of linear-Gaussian policies would be prohibitively expensive when the dimensionality of the optimization problem is high.

Fortunately, many high-dimensional optimization problems have underlying structure that can be exploited. For example, the parameters of neural nets are equivalent up to permutation among certain coordinates. More concretely, for fully connected neural nets, the dimensions of a hidden layer and the corresponding weights can be permuted arbitrarily without changing the function they compute. Because permuting the dimensions of two adjacent layers can permute the weight matrix arbitrarily, an optimization algorithm should be invariant to permutations of the rows and columns of a weight matrix. A reasonable prior to impose is that the algorithm should behave in the same

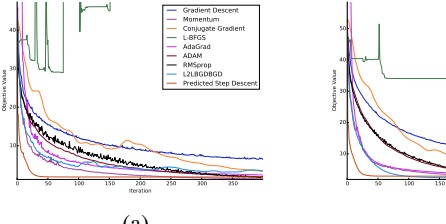 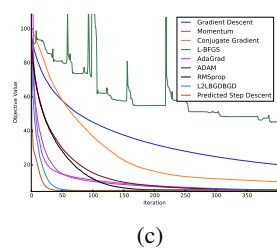

| (a) | (b) | (c) |

Figure 1: Comparison of the various hand-engineered and learned algorithms on training neural nets with 48 input and hidden units on (a) TFD, (b) CIFAR-10 and (c) CIFAR-100 with mini-batches of size 64. The vertical axis is the true objective value and the horizontal axis represents the iteration. Best viewed in colour.

manner on all coordinates that correspond to entries in the same matrix. That is, if the values of two coordinates in all current and past gradients and iterates are identical, then the step vector produced by the algorithm should have identical values in these two coordinates. We will refer to the set of coordinates on which permutation invariance is enforced as a coordinate group. For the purposes of learning an optimization algorithm for neural nets, a natural choice would be to make each coordinate group correspond to a weight matrix or a bias vector. Hence, the total number of coordinate groups is twice the number of layers, which is usually fairly small.

In the case of GPS, we impose this prior on both $\psi$ and $\pi$. For the purposes of updating $\eta$, we first impose a block-diagonal structure on the parameters $A_t, B_t$ and $F_t$ of the fitted transition probability density $\tilde{p}\left(s_{t+1} \mid s_t, a_t, t; \zeta\right) = \mathcal{N}(A_t s_t + B_t a_t + c_t, F_t)$, so that for each coordinate in the optimization problem, the dimensions of $s_{t+1}$ that correspond to the coordinate only depend on the dimensions of $s_t$ and $a_t$ that correspond to the same coordinate. As a result, $\tilde{p}\left(s_{t+1} \mid s_t, a_t, t; \zeta\right)$ decomposes into multiple independent probability densities $\tilde{p}^j\left(s_{t+1}^j \mid s_t^j, a_t^j, t; \zeta^j\right)$, one for each coordinate $j$. Similarly, we also impose a block-diagonal structure on $C_t$ for fitting $\tilde{c}(s_t)$ and on the parameter matrix of the fitted model for $\pi\left(a_t \mid s_t; \theta\right)$. Under these assumptions, $K_t$ and $G_t$ are guaranteed to be block-diagonal as well. Hence, the Bregman divergence penalty term, $D\left(\eta, \theta\right)$ decomposes into a sum of Bregman divergence terms, one for each coordinate.

We then further constrain dual variables $\lambda_t$, sub-vectors of parameter vectors and sub-matrices of parameter matrices corresponding to each coordinate group to be identical across the group. Additionally, we replace the weight $\nu_t$ on $D\left(\eta, \theta\right)$ with an individual weight on each Bregman divergence term for each coordinate group. The problem then decomposes into multiple independent subproblems, one for each coordinate group. Because the dimensionality of the state subspace corresponding to each coordinate is constant, LQG can be executed on each subproblem much more efficiently.

Similarly, for $\pi$, we choose a $\mu_\omega^\pi(\cdot)$ that shares parameters across different coordinates in the same group. We also impose a block-diagonal structure on $\Sigma^\pi$ and constrain the appropriate sub-matrices to share their entries.

### 3.6 FEATURES

We describe the features $\Phi(\cdot)$ and $\Psi(\cdot)$ at time step $t$, which define the state $s_t$ and observation $o_t$ respectively.

Because of the stochasticity of gradients and objective values, the state features $\Phi(\cdot)$ are defined in terms of summary statistics of the history of iterates $\left\{x^{(i)}\right\}_{i=0}^t$, gradients $\left\{\nabla\hat{f}(x^{(i)})\right\}_{i=0}^t$ and objective values $\left\{\hat{f}(x^{(i)})\right\}_{i=0}^t$. We define the following statistics, which we will refer to as the average recent iterate, gradient and objective value respectively:

- $\overline{x^{(i)}} := \frac{1}{\min(i+1,3)} \sum_{j=\max(i-2,0)}^{i} x^{(j)}$

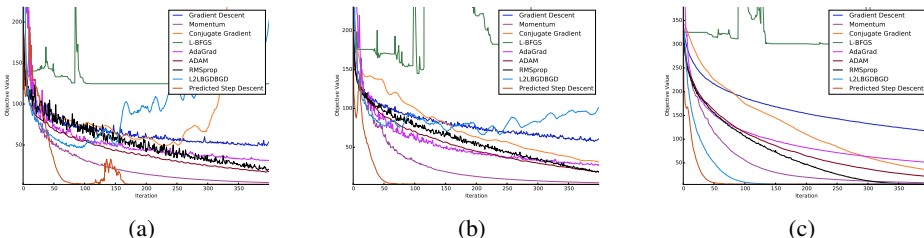

(a)                                    (b)                                    (c)

Figure 2: Comparison of the various hand-engineered and learned algorithms on training neural nets with 100 input units and 200 hidden units on (a) TFD, (b) CIFAR-10 and (c) CIFAR-100 with mini-batches of size 64. The vertical axis is the true objective value and the horizontal axis represents the iteration. Best viewed in colour.

- $\overline{\nabla \hat{f}(x^{(i)})} := \frac{1}{\min(i+1,3)} \sum_{j=\max(i-2,0)}^{i} \nabla \hat{f}(x^{(j)})$

- $\overline{\hat{f}(x^{(i)})} := \frac{1}{\min(i+1,3)} \sum_{j=\max(i-2,0)}^{i} \hat{f}(x^{(j)})$

The state features $\Phi(\cdot)$ consist of the relative change in the average recent objective value, the average recent gradient normalized by the magnitude of the a previous average recent gradient and a previous change in average recent iterate relative to the current change in average recent iterate:

- $\left\{ \left( \overline{\hat{f}(x^{(t-5i)})} - \overline{\hat{f}(x^{(t-5(i+1))})} \right) \big/ \overline{\hat{f}(x^{(t-5(i+1))})} \right\}_{i=0}^{24}$

- $\left\{ \overline{\nabla \hat{f}(x^{(t-5i)})} \big/ \left( \left| \overline{\nabla \hat{f}(x^{(\max(t-5(i+1),t\bmod 5))})} \right| + 1 \right) \right\}_{i=0}^{25}$

- $\left\{ \frac{\left| \overline{x^{(\max(t-5(i+1),t\bmod 5+5))}} - \overline{x^{(\max(t-5(i+2),t\bmod 5))}} \right|}{\left| \overline{x^{(t-5i)}} - \overline{x^{(t-5(i+1))}} \right| + 0.1} \right\}_{i=0}^{24}$

Note that all operations are applied element-wise. Also, whenever a feature becomes undefined (i.e.: when the time step index becomes negative), it is replaced with the all-zeros vector.

Unlike state features, which are only used when training the optimization algorithm, observation features $\Psi(\cdot)$ are used both during training and at test time. Consequently, we use noisier observation features that can be computed more efficiently and require less memory overhead. The observation features consist of the following:

- $\left( \hat{f}(x^{(t)}) - \hat{f}(x^{(t-1)}) \right) \big/ \hat{f}(x^{(t-1)})$

- $\nabla \hat{f}(x^{(t)}) \big/ \left( \left| \nabla \hat{f}(x^{(\max(t-1,0))}) \right| + 1 \right)$

- $\frac{\left| x^{(\max(t-1,1))} - x^{(\max(t-2,0))} \right|}{\left| x^{(t)} - x^{(t-1)} \right| + 0.1}$

## 4 EXPERIMENTS

For clarity, we will refer to training of the optimization algorithm as "meta-training" to differentiate it from base-level training, which will simply be referred to as "training".

We meta-trained an optimization algorithm on a single objective function, which corresponds to the problem of training a two-layer neural net with 48 input units, 48 hidden units and 10 output units on a randomly projected and normalized version of the MNIST training set with dimensionality 48 and unit variance in each dimension. We modelled the optimization algorithm using an recurrent neural net with a single layer of 128 LSTM (Hochreiter & Schmidhuber, 1997) cells. We used a time horizon of 400 iterations and a mini-batch size of 64 for computing stochastic gradients and objective values. We evaluate the optimization algorithm on its ability to generalize to unseen objective functions, which correspond to the problems of training neural nets on different tasks/datasets.

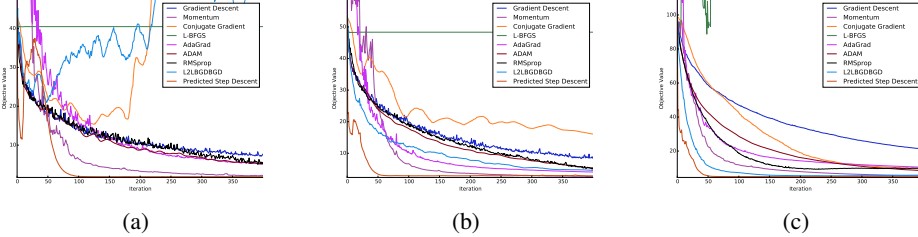

Figure 3: Comparison of the various hand-engineered and learned algorithms on training neural nets with 48 input and hidden units on (a) TFD, (b) CIFAR-10 and (c) CIFAR-100 with mini-batches of size 10. The vertical axis is the true objective value and the horizontal axis represents the iteration. Best viewed in colour.

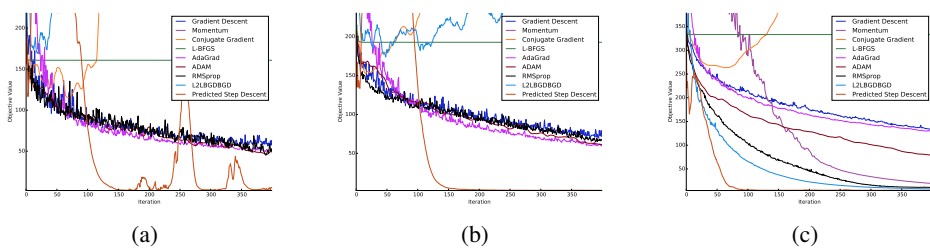

Figure 4: Comparison of the various hand-engineered and learned algorithms on training neural nets with 100 input units and 200 hidden units on (a) TFD, (b) CIFAR-10 and (c) CIFAR-100 with mini-batches of size 10. The vertical axis is the true objective value and the horizontal axis represents the iteration. Best viewed in colour.

We evaluate the learned optimization algorithm on three datasets, the Toronto Faces Dataset (TFD), CIFAR-10 and CIFAR-100. These datasets are chosen for their very different characteristics from MNIST and each other: TFD contains 3300 grayscale images that have relatively little variation and has seven different categories, whereas CIFAR-100 contains 50,000 colour images that have varied appearance and has 100 different categories.

All algorithms are tuned on the training objective function. For hand-engineered algorithms, this entails choosing the best hyperparameters; for learned algorithms, this entails meta-training on the objective function. We compare to the seven hand-engineered algorithms: stochastic gradient descent, momentum, conjugate gradient, L-BFGS, ADAM, AdaGrad and RMSprop. In addition, we compare to an optimization algorithm meta-trained using the method described in (Andrychowicz et al., 2016) on the same training objective function (training two-layer neural net on randomly projected and normalized MNIST) under the same setting (a time horizon of 400 iterations and a mini-batch size of 64).

First, we examine the performance of various optimization algorithms on similar objective functions. The optimization problems under consideration are those for training neural nets that have the same number of input and hidden units (48 and 48) as those used during meta-training. The number of output units varies with the number of categories in each dataset. We use the same mini-batch size as that used during meta-training. As shown in Figure 1, the optimization algorithm meta-trained using our method (which we will refer to as Predicted Step Descent) consistently descends to the optimum the fastest across all datasets. On the other hand, other algorithms are not as consistent and the relative ranking of other algorithms varies by dataset. This suggests that Predicted Step Descent has learned to be robust to variations in the data distributions, despite being trained on only one objective function, which is associated with a very specific data distribution that characterizes MNIST. It is also interesting to note that while the algorithm meta-trained using (Andrychowicz

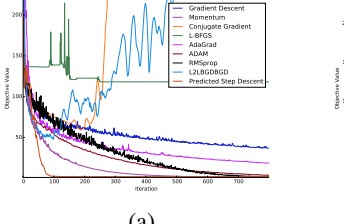 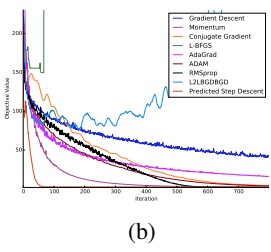 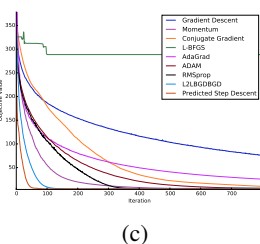

(a) (b) (c)

Figure 5: Comparison of the various hand-engineered and learned algorithms on training neural nets with 100 input units and 200 hidden units on (a) TFD, (b) CIFAR-10 and (c) CIFAR-100 for 800 iterations with mini-batches of size 64. The vertical axis is the true objective value and the horizontal axis represents the iteration. Best viewed in colour.

et al., 2016) (which we will refer to as L2LBGDBGD) performs well on CIFAR, it is unable to reach the optimum on TFD.

Next, we change the architecture of the neural nets and see if Predicted Step Descent generalizes to the new architecture. We increase the number of input units to 100 and the number of hidden units to 200, so that the number of parameters is roughly increased by a factor of 8. As shown in Figure 2, Predicted Step Descent consistently outperforms other algorithms on each dataset, despite having not been trained to optimize neural nets of this architecture. Interestingly, while it exhibited a bit of oscillation initially on TFD and CIFAR-10, it quickly recovered and overtook other algorithms, which is reminiscent of the phenomenon reported in (Li & Malik, 2016) for low-dimensional optimization problems. This suggests that it has learned to detect when it is performing poorly and knows how to change tack accordingly. L2LBGDBGD experienced difficulties on TFD and CIFAR-10 as well, but slowly diverged.

We now investigate how robust Predicted Step Descent is to stochasticity of the gradients. To this end, we take a look at its performance when we reduce the mini-batch size from 64 to 10 on both the original architecture with 48 input and hidden units and the enlarged architecture with 100 input units and 200 hidden units. As shown in Figure 3, on the original architecture, Predicted Step Descent still outperforms all other algorithms and is able to handle the increased stochasticity fairly well. In contrast, conjugate gradient and L2LBGDBGD had some difficulty handling the increased stochasticity on TFD and to a lesser extent, on CIFAR-10. In the former case, both diverged; in the latter case, both were progressing slowly towards the optimum.

On the enlarged architecture, Predicted Step Descent experienced some significant oscillations on TFD and CIFAR-10, but still managed to achieve a much better objective value than all the other algorithms. Many hand-engineered algorithms also experienced much greater oscillations than previously, suggesting that the optimization problems are inherently harder. L2LBGDBGD diverged fairly quickly on these two datasets.

Finally, we try doubling the number of iterations. As shown in Figure 5, despite being trained over a time horizon of 400 iterations, Predicted Step Descent behaves reasonably beyond the number of iterations it is trained for.

## 5 CONCLUSION

In this paper, we presented a new method for learning optimization algorithms for high-dimensional stochastic problems. We applied the method to learning an optimization algorithm for training shallow neural nets. We showed that the algorithm learned using our method on the problem of training a neural net on MNIST generalizes to the problems of training neural nets on unrelated tasks/datasets like the Toronto Faces Dataset, CIFAR-10 and CIFAR-100. We also demonstrated that the learned optimization algorithm is robust to changes in the stochasticity of gradients and the neural net architecture.

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
