# OpenReview forum: "Learning to Optimize Neural Nets"
_ICLR.cc/2018/Conference — Reject_

### Official Review · AnonReviewer2 · 2017-11-24
**See below for details**

**Rating:** 5
**Confidence:** 3

**Review:**

[Main comments]

* I would advice the authors to explain in more details in the intro
what's new compared to Li & Malik (2016) and Andrychowicz et al. (2016).
It took me until section 3.5 to figure it out.

* If I understand correctly, the only new part compared to Li & Malik (2016) is
section 3.5, where block-diagonal structure is imposed on the learned matrices.
Is that correct?

* In the experiments, why not comparing with Li & Malik (2016)? (i.e., without
  block-diagonal structure)

* Please clarify whether the objective value shown in the plots is wrt the training
  set or the test set. Reporting the training objective value makes little
sense to me, unless the time taken to train on MNIST is taken into account in
the comparison.

* Please clarify what are the hyper-parameters of your meta-training algorithm
  and how you chose them.

I will adjust my score based on the answer to these questions.

[Other comments]

* "Given this state of affairs, perhaps it is time for us to start practicing
  what we preach and learn how to learn"

This is in my opinion too casual for a scientific publication...

* "aim to learn what parameter values of the base-level learner are useful
  across a family of related tasks"

If this is essentially multi-task learning, why not calling it so?  "Learning
what to learn" does not mean anything.  I understand that the authors wanted to
have "what", "which" and "how" sections but this is not clear at all.

What is a "base-level learner"? I think it would be useful to define it more
precisely early on.

* I don't see the difference between what is described in Section 2.2
  ("learning which model to learn") and usual machine learning (searching for
the best hypothesis in a hypothesis class).

* Typo: p captures the how -> p captures how

* The L-BFGS results reported in all Figures looked suspicious to me.  How do you
  explain that it converges to a an objective value that is so much worse?
Moreover, the fact that there are huge oscillations makes me think that the
authors are measuring the function value during the line search rather than
that at the end of each iteration.

---

> ### Author Response · Authors · 2018-01-05
> **Response to your review**
>
> The following are new compared to (Li & Malik, 2016):
>
> - A partially observable formulation, which allows the use of observation features that are noisier but can be computed more efficiently than state features. Because only the observation features are used at test time, this improves the time and space efficiency of the learned algorithm.
> - Learns an optimization algorithm that works in a stochastic setting (when we have noisy gradients).
> - Introduced features so that the search is only over algorithms that are invariant to scaling of the objective functions and/or the parameters.
> - The update formula is now parameterized as a recurrent net rather than a feedforward net.
> - The block-diagonal structure on the matrices, which allows the method to scale to high-dimensional problems.
>
> As discussed in Sect. 3.5, the block-diagonal structure is what enables us to learn an optimization algorithm for high-dimensional problems. Because the time complexity of LQG is cubic in the state dimensionality, (Li & Malik, 2016) cannot be tractably applied to the high-dimensional problems considered in our paper.
>
> The objective values shown in the plots are computed on the training set. However, curves on the test set are similar.
>
> Note that the optimization algorithm is only (meta-)trained *once* on the problem of training on MNIST and is *not* retrained on the problems of (base-)training on TFD, CIFAR-10 and CIFAR-100. The time used for meta-training is therefore a one-time upfront cost; it is analogous to the time taken by researchers to devise a new optimization algorithm. For this reason, it does not make sense to include the time used for meta-training when comparing meta-test time performance.
>
> We'll clarify the details on hyperparameters in the camera-ready.
>
> Regarding terminology, "learning what to learn" is a broader area that subsumes multi-task learning and also includes transfer learning and few-shot learning, for example. "Learning which model to learn" is different from the usual base-level learning because the aim is to search over hypothesis classes (model classes) rather than individual hypotheses (model parameters). Note that the use of these terms to refer to multi-task learning and hyperparameter optimization is not some sort of re-branding exercise; it is simply a reflection of how the terms "learning to learn" and "meta-learning" were used historically. For example, Thrun & Pratt's book on "Learning of Learn" (2012) focuses on "learning what to learn", and Brazdil et al.’s book on "Metalearning" (2008) focuses on "learning which model to learn". Because there has never been consensus on the precise definition of "learning to learn", the "what", "which" and "how" subsections in Sect. 2 are simply a convenient taxonomy of the diverse range of methods that all fall under the umbrella of "learning to learn".

---

### Official Review · AnonReviewer3 · 2017-11-28
**This paper proposed a reinforcement learning (RL) based method to learn an optimal optimization algorithm for training shallow neural networks. This work is an extended version of [Li &Malik 2016] aiming to address the high-dimensional problem.**

**Rating:** 6
**Confidence:** 4

**Review:**

This paper proposed a reinforcement learning (RL) based method to learn an optimal optimization algorithm for training shallow neural networks. This work is an extended version of [1], aiming to address the high-dimensional problem.



Strengths:

The proposed method has achieved a better convergence rate in different tasks than all other hand-engineered algorithms.
The proposed method has better robustess in different tasks and different batch size setting.
The invariant of coordinate permutation and the use of block-diagonal structure improve the efficiency of LQG.


Weaknesses:

1. Since the batch size is small in each experiment, it is hard to compare convergence rate within one epoch. More iterations should be taken and the log-scale style figure is suggested.

2. In Figure 1b, L2LBGDBGD converges to a lower objective value, while the other figures are difficult to compare, the convergence value should be reported in all experiments.

3. “The average recent iterate“ described in section 3.6 uses recent 3 iterations to compute the average, the reason to choose “3”, and the effectiveness of different choices should be discussed, as well as the “24” used in state features.

4. Since the block-diagonal structure imposed on A_t, B_t, and F_t, how to choose a proper block size? Or how to figure out a coordinate group?

5. The caption in Figure 1,3, “with 48 input and hidden units” should clarify clearly.
The curves of different methods are suggested to use different lines (e.g., dashed lines) to denote different algorithms rather than colors only.

6. typo: sec 1 parg 5, “current iterate” -> “current iteration”.


Conclusion:

Since RL based framework has been proposed in [1] by Li & Malik, this paper tends to solve the high-dimensional problem. With the new observation of invariant in coordinates permutation in neural networks, this paper imposes the block-diagonal structure in the model to reduce the complexity of LQG algorithm. Sufficient experiment results show that the proposed method has better convergence rate than [1]. But comparing to [1], this paper has limited contribution.

[1]: Ke Li and Jitendra Malik. Learning to optimize. CoRR, abs/1606.01885, 2016.

---

> ### Author Response · Authors · 2018-01-05
> **Response to your review**
>
> The coordinate group depends on the structure of the underlying optimization problem and should correspond to the set of parameters for which the particular ordering among them has little or no significance. For example, for neural nets, the parameters corresponding to the weights in the same layer should be in the same coordinate group, because their ordering can be permuted (by permuting the units above and below) without changing the function the neural net computes.
>
> The inability to scale to high-dimensional problems was actually the main limitation of the previous work (Li & Malik, 2016) [1] – it was unclear at the time if this could be overcome (see for example the reviews of [1] at ICLR 2017). Overcoming the scalability issue therefore represents a significant contribution.

---

### Official Review · AnonReviewer1 · 2017-11-28
**Learning to Optimize Neural Nets**

**Rating:** 6
**Confidence:** 3

**Review:**

Summary of the paper
---------------------------
The paper derives a scheme for learning optimization algorithm for high-dimensional stochastic problems as the one involved in shallow neural nets training. The main motivation is to learn to optimize with the goal to design a meta-learner able to generalize across optimization problems (related to machine learning applications as learning a neural network) sharing the same properties. For this sake, the paper casts the problem into reinforcement learning framework and relies on guided policy search (GPS) to explore the space of states and actions. The states are represented by the iterates, the gradients, the objective function values, derived statistics and features, the actions are the update directions of parameters to be learned. To make the formulated problem tractable, some simplifications are introduced (the policies are restricted to gaussian distributions family, block diagonal structure is imposed on the involved parameters). The mean of the stationary non-linear policy of GPS is modeled as a recurrent network with parameters to be learned. A hatch of how to learn the overall process is presented. Finally experimental evaluations on synthetic or real datasets are conducted to show the effectiveness of the approach.

Comments
-------------
- The overall idea of the paper, learning how to optimize, is very seducing and the experimental evaluations (comparison to normal optimizers and other meta-learners) tend to conclude the proposed method is able to learn the behavior of an optimizer and to generalize to unseen problems.
- Materials of the paper sometimes appear tedious to follow, mainly in sub-sections 3.4 and 3.5. It would be desirable to sum up the overall procedure in an algorithm. Page 5, the term $\omega$ intervening in the definition of the policy $\pi$ is not defined.
- The definitions of the statistics and features (state and observation features) look highly elaborated. Can authors provide more intuition on these precise definitions? How do they impact for instance changing the time range in the definition of $\Phi$) in the performance of the meta-learner?
- Figures 3 and 4 illustrate some oscillations of the proposed approach. Which guarantees do we have that the algorithm will not diverge as L2LBGDBGD does? How long should be the training to ensure a good and stable convergence of the method?
- An interesting experience to be conducted and shown is to train the meta-learner on another dataset (CIFAR for example) and to evaluate its generalization ability on the other sets to emphasize the effectiveness of the method.

---

> ### Author Response · Authors · 2018-01-05
> **Response to your review**
>
> Below is an intuitive explanation of the state and observation features:
>
> Average recent iterate, gradient and objective value are the means over the three most recent iterates, gradients and objective values respectively, unless there are fewer than three iterations in total, in which case the mean is taken over the iterations that have taken place so far.
>
> The state features consist of the following:
> - The relative change in the average recent objective value compared to five iterations ago, as of every fifth iteration in the 120 most recent iterations; intuitively, this can capture if and by how much the objective value is getting better or worse.
> - The average recent gradient normalized by the element-wise magnitude of the average recent gradient five iterations ago, as of every fifth iteration in the 125 most recent iterations.
> - The normalized absolute change in the average iterate from five iterations ago, as of every fifth iteration in the 125 most recent iterations; intuitively, this can capture the per-coordinate step sizes we used previously.
>
> Similarly, the observation features consist of the following:
> - The relative change in the objective value compared to the previous iteration
> - The gradient normalized by the element-wise magnitude of the gradient from the previous iteration
> - The normalized absolute change in the iterate from the previous iteration
>
> The normalization is designed so that the features are invariant to scaling of the objective function and to reparameterizations that involve scaling of the individual parameters.
>
> The reason that the algorithm learned using the proposed approach does not diverge as L2LBGDBGD does is because the training is done under a more challenging and realistic setting, namely when the local geometries of the objective function are not known a priori. This is the setting under which the learned algorithm must operate at test time, since the geometry of an unseen objective function is unknown. This is the key difference between the proposed method and L2LBGDBGD, and more broadly, between reinforcement learning and supervised learning. L2LBGDBGD assumes the local geometry of the objective function to be known and so requires the local geometries of the objective function seen at test time to match the local geometries of one of the objective functions seen during training. Whenever this does not hold, it diverges. As a result, there is very little generalization to different objective functions. On the other hand, the proposed approach does not assume known geometry and therefore the algorithm it learns is more robust to differences in geometry at test time.
>
> In reinforcement learning (RL) terminology, L2LBGDBGD assumes that the model/dynamics is known, whereas the proposed method assumes the model/dynamics is unknown. In the context of learning optimization algorithms, the dynamics captures what the next gradient is likely to be given the current gradient and step vector, or in other words, the local geometry of the objective function.
>
> The reason why the algorithm learned using the proposed approach oscillates in Figs. 3 and 4 is because the batch size is reduced to 10 from 64 (which was the batch size used during meta-training), and so the gradients are noisier. Importantly, the algorithm is able to recover from the oscillations and converge to a good optimum in the end, demonstrating the robustness of the algorithm learned using the proposed approach.
>
> In practice, about 10-20 iterations of the GPS algorithm are needed to obtain a good optimization algorithm.

---

### Decision · Program_Chairs · 2018-01-29
**ICLR 2018 Conference Acceptance Decision**

**Decision:**

Reject

**Comment:**

The presented work is a good attempt to expand the work of Li and Malik to the high-dimensional, stochastic setting. Given the reviewer comments, I think the paper would benefit from highlighting the comparatively novel aspects, and in particular doing so earlier in the paper.

It is very important, given the nature of this work, to articulate how the hyperparameters of the learned optimizers, and of the hand-engineered optimizers are chosen. It is also important to ensure that the amount of time spent on each is roughly equal in order to facilitate an apples-to-apples comparison.

The chosen architectures are still quite small compared to today's standards. It would be informative to see how the learned optimizers compare on realistic architectures, at least to see the performance gap.

Please clarify the objective being optimized, and it would be useful to report test error.

The approach is interesting, but does not yet meet the threshold required for acceptance.